# Vascular Aging and Damage in Patients with Iron Metabolism Disorders

**DOI:** 10.3390/diagnostics12112817

**Published:** 2022-11-16

**Authors:** Krzysztof Młodziński, Michał Świątczak, Justyna Rohun, Jacek Wolf, Krzysztof Narkiewicz, Marcin Hellmann, Ludmiła Daniłowicz-Szymanowicz

**Affiliations:** 1II Department of Cardiology and Electrotherapy, Medical University of Gdańsk, Dębinki 7, 80-211 Gdańsk, Poland; 2Department of Hypertension and Diabetology, Medical University of Gdańsk, Dębinki 7, 80-211 Gdańsk, Poland; 3Department of Cardiac Diagnostics, Medical University of Gdańsk, Dębinki 7, 80-211 Gdańsk, Poland

**Keywords:** vascular aging, iron metabolism, iron overload, iron deficiency

## Abstract

Vascular aging is a physiological, multifactorial process that involves every type of vessel, from large arteries to microcirculation. This manifests itself as impaired vasomotor function, altered secretory phenotype, deteriorated intercellular transport function, structural remodeling, and aggravated barrier function between the blood and the vascular smooth muscle layer. Iron disorders, particularly iron overload, may lead to oxidative stress and, among other effects, vascular aging. The elevated transferrin saturation and serum iron levels observed in iron overload lead to the formation of a non-transferrin-bound iron (NTBI) fraction with high pro-oxidant activity. NTBI can induce the production of reactive oxygen species (ROS), which induce lipid peroxidation and mediate iron-related damage as the elements of oxidative stress in many tissues, including heart and vessels’ mitochondria. However, the available data make it difficult to precisely determine the impact of iron metabolism disorders on vascular aging; therefore, the relationship requires further investigation. Our study aims to present the current state of knowledge on vascular aging in patients with deteriorated iron metabolism.

## 1. Introduction

Cardiovascular diseases (CVD) are the most common cause of death worldwide [1]. Among the factors leading to CVD, aging is the most important one [2]. However, the exact mechanisms of that process have not been comprehensively evaluated. Among others, a decrease in compliance and arteries’ stiffening are the hallmarks of vascular aging [3]. A lifetime exposure to a number of cardiovascular risk factors translates to an increased risk for organ damage in elderly individuals [4]. Iron disorders, particularly iron overload, have been documented to accelerate vascular aging [5]. The pro-oxidant properties of iron promote free radicals’ formation, which in turn facilitates early vascular aging [6,7]. The discovery of the possible mechanisms responsible for this process and further new diagnostic methods could allow for the detection of such pathologies early on and implementation actions to delay the development of the potential complications of CVD. The present article aims to outline the current state of knowledge with respect to vascular aging in patients with iron metabolism disorders.

## 2. Materials and Methods

In preparing the content of our manuscript, we followed the Preferred Reporting Items Requirements for Systematic Reviews and Meta-Analysis (PRISMA 2020) guidelines [8]. Relevant research studies were sourced using PubMed, Scopus, and Wiley electronic databases. Databases were searched using the following keywords: endothelial function hemochromatosis, endothelial function iron overload and the keywords pulse wave velocity, microvascular function, skin microcirculation, flow-mediated skin fluorescence, and laser speckle contrast imaging, combined with the statements, iron, iron overload, and hemochromatosis. Through database scanning, a total of 1918 works were identified. Duplicates (126 works) were manually eliminated. Publications were screened using the following filters: English language and publication date, i.e., articles published between 1957 and 2022. Then, we performed a preliminary selection of the remaining articles based on the inclusion criteria: only studies on iron metabolism problems connected to the topic of vascular damage were considered. We screened titles first, then the abstracts, and, only where necessary (i.e., the issue was not evident from the title and/or abstract reading), the authors conducted a full-text evaluation. A total of 1681 articles were eliminated due to irrelevance after inspecting the title and abstract. Eventually, 109 papers were selected for use in the article creation process, including 59 original articles, 49 review papers, and 1 brief communication; 96 of them were related to cardiological issues (Figure 1).

## 3. The Pathophysiology of Vascular Aging

Vascular aging is a physiological multifactorial process that involves every type of vessel, from large arteries to microcirculation. The factors responsible for this effect can be divided into molecular, which primarily include oxidative stress, genome instability, or epigenetic modifications, and cellular factors, which include mitochondrial dysfunction, cellular senescence, or impaired endothelial function [9]. Vascular aging manifests itself as impaired vasomotor function, altered secretory phenotypes, impaired intercellular transport function, structural remodeling, or impaired barrier function between the blood and the vascular smooth muscle layer [10]. The changes observed during endothelial aging include multiple factors, such as the disruption of the integrity of the cytoskeleton, the impaired production of endothelial NO, or an increased release of endothelin 1. Endothelin-1 is a vasoconstrictor and increases the expression of the adhesion molecules VCAM-1 and ICAM-1, which leads to increased cell adhesion, including monocytes. In addition, there is also an increase in the expression of the NF-kB factor. It further promotes the inflammatory response and the release of several cytokines, such as tumor necrosis factor-a (TNFα), interleukin-6 (IL-6), monocyte chemotactic protein-1, and adhesion molecules [11]. Ultimately, an increased susceptibility of endothelial cells (ECs) to apoptosis is observed [12].

Vascular aging is significantly influenced by oxidative stress. This hypothesis was first put forward by Harman in the 1950s, who stated that the accumulation of ROS is a major cause of the sequential changes that characterize advancing age and the progressive increase in disease and death [13]. ROS in the vessel walls act as mediators, influencing the modulation of the immune response, or the structure and survival of endothelial and vascular smooth muscle cells (VSMCs); however, the abnormal control of these processes leads to mitochondrial dysfunction and increased ROS production [14,15]. Excess oxygen free radicals lead to the uncoupling of endothelial nitric oxide synthase, resulting in the formation of nitric oxide peroxide and the decreased expression of coenzyme tetrahydrobiopterin (BH4) [16]. BH4 is a cofactor for various enzymes, including nitric oxide synthase. The current data suggest that reduced bioavailability of BH4 in blood vessels is one mechanism for the development of impaired endothelial NO bioavailability/synthesis in a variety of disease states, such as diabetes, hypertension, hypercholesterolemia, arteriosclerosis, and aging [16]. This compound reacting with nitric oxide leads to the formation of the peroxynitrite anion, which is characteristic of aging ECs [7]. It causes direct cytotoxic effects, adverse effects on mitochondrial function, and the activation of inflammatory pathways, which are associated with the impaired regulation of blood vessel diameter, increased vasoconstriction, and impaired tissue perfusion [17]. In a study conducted on rats, researchers observed that despite a decrease in the expression of endothelial nitric oxide synthase, there was an increase in NO levels, which was due to an increased expression of inducible NO synthase [15]. Coexisting oxidative stress and an incremented expression of inducible nitric oxide synthase led to the increased production of the peroxynitrite anion, which led to increasing endothelial dysfunction manifested as vasoconstriction. It is also noteworthy that NO exerts potent anti-inflammatory and anticoagulant effects and prevents leukocyte adhesion, so impaired endothelial NO production likely promotes a proatherogenic vascular phenotype in aging [18]. Increased vascular oxidative stress has also been linked to the activation of matrix metalloproteinases (MMPs) and the resulting disruption of the structural integrity of aging arteries, potentially contributing to the stiffening of large arteries and the pathogenesis of CVDs such as coronary artery disease (CAD), arterial hypertension, or atherosclerosis [19]. 

## 4. Diagnostic Tools of Vascular Damage

With the increasing availability of various diagnostic-imaging techniques, it could be possible to visualize both large and smallest blood vessels. Among the methods that allow for the determination of large- and medium-caliber vascular damage are the evaluation of the pulse wave velocity, the vasodilatation of the brachial artery after ischemia, or the assessment of the intima-media complex [20,21,22]. Microcirculation diagnostics, capillaroscopy, laser Doppler techniques, laser speckle contrast imaging, or optical coherence tomography angiography are distinguished [23,24,25]. Table 1 provides a complex overview of the techniques used in the diagnosis of vascular aging according to vascular diameter. The table also includes the sensitivity and specificity values available in the literature for the selected diagnostic methods used in the evaluation of vascular damage.

## 5. Clinical Consequences of Vascular Aging

Vascular aging promotes numerous clinical complications including hypertension, myocardial infarction, ischemic and hemorrhagic strokes, aneurysms in large arteries, vascular dementia, macular degeneration, and an increased risk of developing of Alzheimer’s disease [15]. Table 2 summarizes the clinical complications of vascular aging.

## 6. Systemic and Cell Regulation of Iron Metabolism

Iron is one of the essential micronutrients for the functioning of the physiological body. The body’s leading sources of iron include heme compounds (such as hemoglobin or myoglobin), heme enzymes, and non-heme compounds, which are complex forms of iron attached to proteins such as flavin-iron enzymes, transferrin, and ferritin [69]. Heme iron is found in essential proteins such as hemoglobin and myoglobin. Iron, as a component of hemoglobin, mediates the binding and transport of oxygen to cells. Iron ions in myoglobin enable oxygen storage in striated muscle tissue, allowing for efficient muscle action with decreased oxygen partial pressure in the muscles. Nearly two-thirds of the body’s iron is present in circulating erythrocytes as a part of hemoglobin, 30% is present in a store of iron that can be easily accessed, and the remaining 10% is bound to myoglobin in muscle tissue and several enzymes involved in oxidative metabolism and numerous other cell functions [70]. As mentioned earlier, iron directly or indirectly regulates many processes in the body, such as erythropoietin synthesis, bone marrow erythrocyte production, the production of numerous neurotransmitters and enzymes, and the activation and proliferation of lymphocytes.

Due to iron’s toxic properties, the iron level must be strictly regulated in the human body. The regulation of the body’s iron concentration at the systemic level is complex and is based on the activity of the peptide hormone hepcidin and protein transferrin (Tf) [71,72]. Initially, iron is absorbed from the duodenum and small intestine into enterocytes and macrophages, which combine with apoferritin to form ferritin. Iron in this form, with the participation of the protein ferroportin (FPN), is then transported further into the bloodstream [73]. During the transport of iron in the form of ferritin with the participation of FPN, the Fe^2+^ ion is detached and enters the bloodstream. At the same time, apoferritin remains in the cells, which binds further Fe^2+^ ions [74]. Once iron is released from ferritin, it is captured by Tf present in the blood serum. By binding to the receptor for transferrin (sTfR), this protein enables the direct delivery of iron from its stores to the bone marrow. An increase in hepcidin activity causes it to bind to FPN, internalize FPN, and thereby block further iron transport from macrophages and enterocytes [75]. Hepcidin activity depends on stimuli such as elevated iron levels or current inflammation, which contribute to the induction of hepcidin synthesis [75]. In contrast, increased erythropoiesis, iron deficiency, and hypoxia decrease the activity of this hormone. Transferrin production depends mainly on the iron concentration.

The presence of the FPN protein on the cellular surface makes hepcidin responsible for both systemic and cellular-level iron regulation. At the cellular level, iron concentration regulation is controlled post-transcriptionally with the involvement of iron-regulatory proteins.

Despite the existence of complex processes regulating iron metabolism, there is a fraction of iron, termed labile plasma iron (LPI), which is translocated across cell membranes in an unregulated manner and can lead to excessive iron accumulation in the liver, heart, pancreas, and other endocrine organs [71]. This can lead to iron overload and the formation of the non-transferrin-bound iron (NTBI) fraction, which, like LPI, has high pro-oxidant activity [76].

## 7. Iron Metabolism Disorders as the Trigger of Oxidative Stress

Iron disorders, in particular iron overload, which is characterized by elevated transferrin saturation and baseline plasma iron concentration, are associated with increasing NTBI fractions. NTBI, as an extremely toxic iron form, plays a key role in the pathogenesis of iron-related damage in various conditions characterized by iron overload [7]. NTBI fractions are formed when the binding capacity of hemoglobin and the heme released during hemolysis exceeds that of haptoglobin (Hp) and hemopexin (Hx), resulting in the formation of non-haptoglobin-bound hemoglobin (NHBHB) and hemopexin-bound heme (NHBH) [77]. While hemoglobin–haptoglobin and heme–hemopexin complexes are taken up by macrophages and hepatocytes, respectively, NHBHB and NHBH freely enter cells in a non-specific manner, causing heme overload and tissue damage [78]. NTBI can induce the production of reactive oxygen species (ROS), which increase lipid peroxidation and mediate iron-related damage in many tissues, including vessels and heart mitochondria [71]. High levels of NTBI have been found not only in disorders associated with iron overloads, such as hereditary hemochromatosis and beta thalassemia, but also in diseases originally unrelated to iron overload, including diabetes, myelodysplastic syndromes, and end-stage renal disease [7].

NTBI is a source of iron ions, which at first readily undergo the Haber–Weiss reaction and the Fenton reaction, eventually leading to the formation of oxygen free radicals [7]. These molecules contribute to the acceleration of vascular aging through the peroxidation of lipids and proteins in cell membranes and direct effects on ECs, VSMCs, and macrophages. The peroxidation of polyunsaturated fatty acids leads to the formation of byproducts such as reactive aldehydes or gamma-keto aldehydes, which are likely responsible for the vasotoxic and pro-inflammatory mechanisms [79]. In turn, ECs, due to NTBI exposure, respond by increasing the expression of adhesion proteins. VCAM-1, or ICAM-1 adhesion proteins together with chemotactic factors, increase the production of intracellular oxygen free radicals, which in turn leads to their death.

Moreover, NTBI contributes to the uncoupling of endothelial nitric oxide synthase, thereby reducing NO availability and causing the impairment of endothelium-dependent vasodilatory function [80]. VSMCs lack the protein ferroportin and, therefore, in the case of iron overload, they accumulate NTBI, resulting in an overproduction of oxygen free radicals, followed by the apoptosis of these cells mediated by monocyte chemotactic factor 1 [81]. This is associated with iron accumulation in the vessel walls and is much more pronounced in patients with a content of high plasma ferritin, in whom iron is strongly deposited in VSMCs of the aortic middle layer [82]. It has been observed that the exposure of VSMCs to NTBI successively leads to their iron overload, the production of oxygen free radicals, and, ultimately, the apoptosis of these cells. In turn, the apoptosis of VSMCs stimulates the recruitment of monocytes into the growing atherosclerotic plaque, thereby promoting its enlargement [83]. Animal studies with an accompanying iron overload are demonstrating that this condition leads to increased aortic stiffness due to increased collagen deposition in the wall [84]. Eventually, when macrophages are exposed to NTBI, they undergo a phenotypic change with an increase in the number of M1-type macrophages, which are characterized by pro-inflammatory activity, producing numerous interleukins and pro-inflammatory cytokines [85].

There are apparent differences in iron metabolism disorders according to sex. Although it affects men and women, the frequency is different [86,87]. Women in the reproductive stage have lower total body iron stores than men because of decreased testosterone levels and ongoing blood loss during menstruation [88]. Postmenopausal women, in contrast to reproductive women, experience menstrual arrest as a result of reduced estrogen levels brought on by the suppression of ovulation, which leads to a rise in serum iron levels, ferritin levels, and transferrin saturation [89]. As a result, postmenopausal women are more likely than women in their childbearing years to experience the side effects of iron overload [71].

### 7.1. Iron Overload Pathologies

Oxidative stress caused by an iron overload status can lead to vascular aging. As mentioned above, an increased serum iron concentration and transferrin saturation lead to an increased formation of NTBI with high pro-oxidant activity. Consequently, NTBI, being a source of iron ions, provides the substrates necessary for the reaction governing the formation of oxygen free radicals and ultimately induces oxidative stress. Additionally, iron overload causes organ damage through ROS by promoting cell senescence, which can lead to liver damage, diabetes, cardiac dysfunction, the dysfunction of the endocrine system, or ocular disease [71]. As a result, vascular aging is accelerated in two different ways. First of all, iron overload is connected with an increase in the pool of free iron. The free iron pool eventually influences the process of vascular aging by acting as a substrate for oxygen free radical production processes. ROS cause vascular endothelium damage and ultimately accelerate vascular aging through several mechanisms outlined above. Secondly, insulin resistance, hyperglycemia, and concomitant inflammation can result in micro- and macrovascular alterations such as retinopathy, CAD, and nephropathy in type 2 diabetes, which occurs as a consequence of iron overload. The aforementioned pathophysiological changes in arteries result in the thickening of the basement membrane, the increased permeability of endothelial cells, and a higher risk of developing an aneurysm, which ultimately accelerates the vascular aging process [90,91]. In experimental studies on mice models of diabetes with iron overload, iron also hastens the development of diabetic retinopathy and nephropathy, which accelerates internal organ damage during diabetes [92]. In human studies, type 2 diabetes, developed as a consequence of hemochromatosis, exhibits evidence of accelerated vascular aging [90]. For instance, Peterlin et al. demonstrated that a C282Y mutation was an independent risk factor for developing diabetic retinopathy [93]. Other authors [94,95] documented that the presence of an HFE gene mutation increases the risk of diabetic nephropathy. Moczulski et al. demonstrated that carriers of mutations in alleles of the H63D gene experience this complication significantly more frequently than non-carriers [94]. Olivia et al. found that the increased risk of developing diabetic nephropathy was associated with carrying at least one mutation in both alleles of the C282Y gene or both alleles of the H63D gene [95].

Data from the literature regarding iron overload and vascular aging are sparse, and are usually based on single studies conducted on small groups of individuals, which makes it impossible to draw valuable conclusions. Hereditary hemochromatosis (HH), beta thalassemia, and sickle cell anemia are the most prominent diseases associated with iron overload, which can affect the aging of blood vessels.

#### 7.1.1. Hemochromatosis

Hereditary hemochromatosis (HH) is one of the most common inherited metabolic diseases and it is associated with an 80% mutation of the HFE gene at position C282Y [71,72]. It leads to impaired iron metabolism through the increased absorption of this micronutrient from the gastrointestinal tract and its abnormal distribution. In HH, an increased level of LPI is observed, which, as with NTBI, undergoes a Fenton reaction, leading to increased ROS production and subsequent oxidative stress [84]. The literature data regarding the influence of HH on vessel function are scarce [96,97]. For example, Gaenzer et al. conducted a study of 119 men and women suffering from HH [96], in which an association between iron overload and endothelial dysfunction was determined by endothelium-dependent vasodilation (EDD) and a measurement of intima thickness [96]. EDD is flow-dependent vasodilation following an increase in shear stress on the arterial endothelium during reactive congestion after a period of ischemia [96]. In this study, the authors demonstrated that among male patients who were not treated with phlebotomies, the EDD parameter was significantly reduced and intima thickness was increased contrary to the control group and HH patients treated with phlebotomies. The initiation of phlebotomies in these patients was associated with significant improvements regarding both iron metabolism (by a reduction in the parameters of iron overload) and the EDD parameter. Furthermore, among men treated with phlebotomies from the beginning of the study, no reduction in EDD was observed. Among the studied women, there were no statistically significant differences in the parameters assessed between the groups of women before and after phlebotomies [96]. In their study, Failla et al. performed on only 12 patients with hemochromatosis and observed a significantly greater radial artery wall thickness in these patients compared to healthy controls; however, the differences in vasodilatation were not statistically significant [97]. Moreover, a significant reduction in wall thickness and a significant increase in vasodilatation were observed after the reduction in the iron level. The data mentioned above may suggest that HH, through oxidative stress, can cause vascular damage and, subsequently, accelerate vascular senescence. However, this needs to be evaluated in further studies.

#### 7.1.2. Beta Thalassemia

Beta thalassemia is an autosomal recessive disease genetic caused by a mutation of genes located on chromosome 11 [98]. It presents with hemolytic anemia caused by abnormalities in the synthesis of beta chains of hemoglobin [98]. Despite the presence of anemia, which is caused by ineffective erythropoiesis, elevated iron and ferritin levels are observed [99]. Such disruptions of iron metabolism are caused by the necessity of life-long blood transfusions in the patient [99]. Kukongviriyapan et al., while conducting a study among 22 patients suffering from beta thalassemia, proposed that the auto-oxidation of excess beta globin chains and iron accumulation leads to oxidative stress and subsequent tissue and organ damage [100]. The authors suggested that the impaired endothelial function in these patients is a result of the ROS-mediated inactivation of endothelial NO synthase [101]. Although no reduction in NO levels was observed among the patients studied, the biological activity of this enzyme was significantly reduced [100]. This may imply that endothelial damage among beta thalassemia patients is caused by the release of hemoglobin from erythrocytes, which undergo auto-oxidation leading to the formation of ROS and thus play a role in endothelial dysfunction. However, further research should be performed for a precise investigation of that process.

#### 7.1.3. Sickle Cell Disease (SCD)

Sickle cell anemia is an autosomal dominant hemoglobinopathy that is associated with the presence of an abnormal sickle hemoglobin structure (HbS) [102]. Erythrocytes in this disease are characterized by a change in shape, impaired deformability, and increased adhesion to the endothelium [102]. This is associated with intra- and extravascular hemolysis and the resulting release of free hemoglobin. The hemoglobin released from the hemolyzed erythrocytes contributes to the removal of endothelium-produced NO, leading to regional NO deficiency [103]. This reaction already occurs at low concentrations of free hemoglobin on the order of 6–10 micromoles, resulting in regional vasoconstriction [104]. In addition, the heme released from hemoglobin leads to the activation of TLR4 and the inflammasome, which are responsible for the development of the inflammatory process [105]. Another factor that impairs endothelial function is the inhibition of NO-dependent vasodilators by free hemoglobin. Ultimately, hemolysis leads to the release of the enzyme arginase-1 from erythrocytes, which metabolizes arginine to ornithine, reducing the amount of available substrate for nitric oxide synthase [103]. As a result, nitric oxide synthase begins to produce superoxide instead of NO and this leads to the formation of peroxynitrite anions that contribute to the induction of oxidative stress, vascular endothelial damage, and, thus, vascular aging [106]. Available evidence in the literature is insufficient for drawing firm conclusions about the impact of SCD on vascular aging. This issue requires further research.

### 7.2. Iron Deficiency

Currently, there is a lack of work in the literature addressing the implications of iron deficiency on vascular aging among people. Dong et al., in their experimental study conducted on rats, showed that inducing iron deficiency leads to an increased production of nitrotyrosine protein, expression of caveolin-1, and the p47phox subunit of NADPH oxidase in the hearts of these animals [107]. The increased expression of a particular NADPH oxidase subunit can result in an increased production of oxygen free radicals, which, by reacting with NO, leads to the formation of peroxynitrite anions, which can cause oxidative stress and ultimately endothelial dysfunction [108]. It is worth noting that nitrotyrosine is a marker of the presence of peroxynitrite anions in the body [109]. This may suggest that increased nitrotyrosine levels are associated with an increased production of ROS and, thus, the presence of oxidative stress. On the other hand, the caveolin-1 protein regulates NO-dependent cell signaling by inhibiting endothelial nitric oxide synthase, which in turn can lead to impaired vasodilatory, antithrombotic, and anti-inflammatory functions, as well as the formation of peroxynitrite anions, which, through oxidative stress, can also lead to impaired vascular endothelial function [107]. This data may suggest a link between iron deficiency and vascular degeneration. Concerning iron deficiency, no increase in NTBI and LPI fractions is observed, which means that endothelial damage and the consequent acceleration of vascular senescence are most likely not caused by oxygen free radicals formed in the Fenton reaction [107]. However, further studies are required to conclusively determine the correlation between vascular aging and iron deficiency.

## 8. Conclusions

Aging is one of the most important risk factors for developing CVD. This process involves vessels of each caliber, ranging from large arteries to capillaries, and leads to a deterioration of the susceptibility and increased stiffness of arterial walls. Modern diagnostic methods, which enable imaging of progressively smaller vessels, may facilitate the diagnosis of vascular damage at an early stage. Vascular aging is a complex and multifactorial process. Among the potential risk factors are iron metabolism disorders, in particular iron overload, which through oxidative stress, the impaired vasodilatory function of the vascular endothelium, and pro-inflammatory effects may lead to an acceleration in this process. However, the data on this issue are insufficient to precisely determine the impact of iron metabolism disorders on vascular senescence; therefore, this topic requires further research.

## Figures and Tables

**Figure 1 diagnostics-12-02817-f001:**
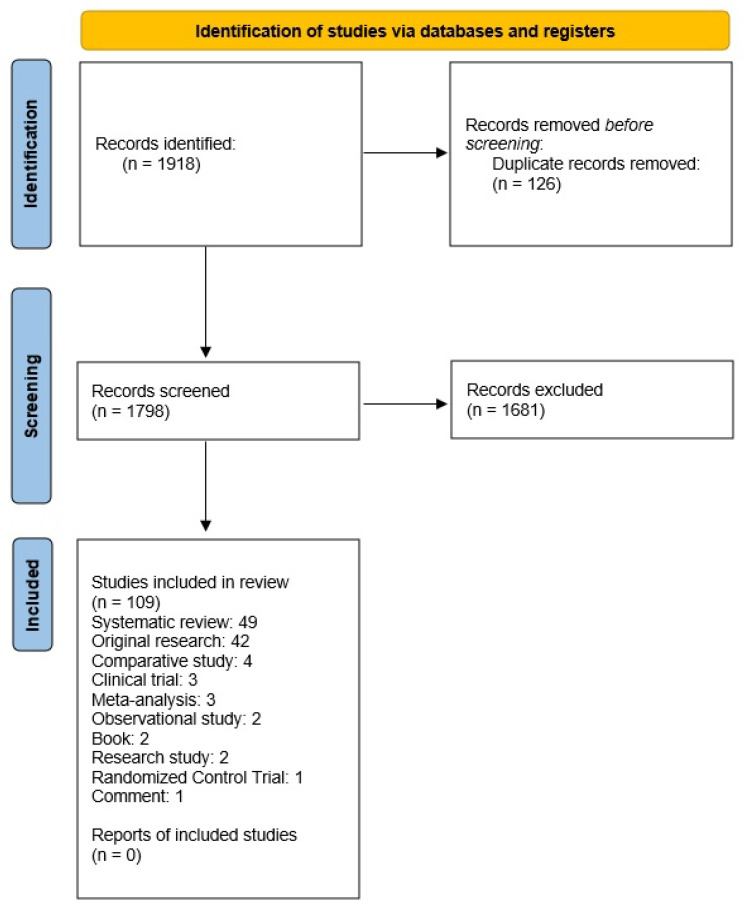
A flow diagram following the PRISMA template. PRISMA preferred reporting items for systematic reviews and meta-analyses.

**Table 1 diagnostics-12-02817-t001:** Diagnostic tools of vascular aging in dependence on vessel diameter.

Imaging Techniques	Description	Vessel Diameter That Can Be Examined	Sensitivity	Specificity
Pulse wave velocity (PWV)	Measures the velocity of arterial pressure waves along the aorta and large arteries; it is usually calculated by dividing the distance by the time of passage of the pressure wave at two points of arterial recording [26]. Depending on the caliber of the vessels, PWV can be divided into carotid-femoral (cfPWV) and brachial-femoral (baPWV) [20].	Large arteries (carotid, brachial, and femoral arteries)	83% (for assessing cardiovascular mortality using cfPWV)—single study [20] 57.1% (for assessing cardiovascular mortality using baPWV)—single study [20] 62% (for detection of the presence of atherosclerotic alterations)—single study [27] 60% (for detection of the patients without atherosclerotic alterations and with high 10-year cardiovascular mortality risk)—single study [27] 83% (for detection of coronary microcirculatory disfunction)—single study [28] 93% (for predicting cardiovascular mortality)—single study [29] 72% (for predicting overall mortality)—single study [29] 34.3% (for detection of people with moderate risk of CVD)—meta-analysis [30] 57.2% (for detection of people with high risk of CVD)—meta-analysis [30]	71% (for assessing cardiovascular mortality using cfPWV–single study [20] 93.1% (for assessing cardiovascular mortality using baPWV)—single study [20] 67% (for detection of the presence of atherosclerotic alterations)—single study [27] 84% (for detection of patients without atherosclerotic alterations and with high 10-year cardiovascular mortality risk)—single study [27] 82% (for detection of coronary microcirculatory disfunction)—single study [28] 60% (for predicting cardiovascular mortality)—single study [29] 62% (for predicting overall mortality)—single study [29] 95.3% (for detection of people with moderate risk of CVD)—meta-analysis [30] 95.3% (for detection of people with high risk of CVD)—meta-analysis [30]
Flow-mediated dilatation (FMD)	Relies on the evaluation of endothelial function in response to ischemia [31]. This test can be performed at various sites in the peripheral circulation. Arterial dilatation is defined as the percentage change in vessel diameter caused by ischemia relative to the vessel diameter before ischemia [31].	Large arteries (carotid, brachial, and femoral arteries)	95% (for excluding CAD)—single study [32] 53% (for assessing the presence of CAD using low flow FMD)—single study [33] 53% (for assessing the presence of CAD using FMD)—single study [33] 44% (for assessing the presence of CAD using low flow FMD and FMD)—single study [33]	60% (for excluding CAD)—single study [32] 80% (for assessing the presence of CAD using low flow FMD)—single study [33] 69% (for assessing the presence of CAD using FMD)—single study [33] 85% (for assessing the presence of CAD using low flow FMD and FMD)—single study [33]
Assessment of the intima-media complex (IMT)	Measures the thickness of the middle layer (tunica intima) and an inner layer (tunica media), that is, the two innermost layers of the arterial wall [34]. The measurement is usually performed with external ultrasound, and sometimes with internal, invasive ultrasound catheters [34].	Large arteries (carotid, femoral, radial, brachial arteries)	66% (for diagnosing CAD)—meta-analysis [22] 43% (for the prediction of CAD)—single study [35] 68.1% (for predicting the prevalence of CVD based on IMT assessment in internal carotid artery)—single study [36] 63.5% (for predicting the prevalence of CVD based on IMT assessment in common carotid artery)—single study [36] 70% (for the diagnosis of CAD)—single study [37]	79% (for diagnosing CAD)—meta-analysis [22] 77% (for the prediction of CAD)—single study [35] 72.3% (for predicting the prevalence of CVD based on IMT assessment in internal carotid artery)—single study [36] 65.4% (for predicting the prevalence of CVD based on IMT assessment in common carotid artery)—single study [36] 75% (for the diagnosis of CAD)—single study [37]
Capillaroscopy	A non-invasive diagnostic method based on the evaluation of capillary morphology in the skin and mucous membranes [38]. It involves viewing capillaries under a special microscop, after moistening the examined area with fluid [38].	Capillaries (the most comprehensive application is the evaluation of the nail shaft vessels)	100% (for the diagnosis of circulating endothelial cells)—single study [39] 89.5% (for diagnosis of scleroderma pattern in patients with systemic sclerosis)—single study [40] 33.3% (for diagnosis of systemic lupus erythematosus pattern in patients with SLE)—single study [40] 60% (for diagnosis of polymyositis/dermatomyositis pattern in patients with PM/DM)—single study [40]	90% (for the diagnosis of circulating endothelial cells)—single study [39] 80% (for diagnosis of scleroderma pattern in patients with systemic sclerosis)—single study [40] 94.4% (for diagnosis of systemic lupus erythematosus pattern in patients with SLE)—single study [40] 96.3% (for diagnosis of polymyositis/dermatomyositis pattern in patients with PM/DM)—single study [40]
Laser-Doppler Flowmetry	Enables the assessment of microvascular function by quantifying rapid changes in cutaneous blood flow in response to a given pharmacological or mechanical stimulus [41]. It has been shown that peripheral endothelial dysfunction of peripheral vessels correlates with endothelial function in coronary microvessels, suggesting that endothelial dysfunction is a globalized pathological condition [41].	Microvascular perfusion	82% (for identification of patients with coronary artery disease)—single study [42] 96% (for differentiating lower limbs with normal microcirculation from ischemic lower limbs)—single study [43]	97% (for identification of patients with coronary artery disease)—single study [42] 96% (for differentiating lower limbs with normal microcirculation from ischemic lower limbs)—single study [43]
Laser speckle contrast imaging (LSCI)	LSCI allows for non-invasive real-time monitoring of peripheral microcirculatory perfusion on a wide area of tissue with a very good spatial and temporal resolution and excellent reproducibility [44,45]. This technique coupled with vascular reactivity tests enables the assessment of endothelial function [46].	Microvascular perfusion	65% (for assessment of endothelial dysfunction in patients with type I diabetes)—single study [47]	87.5% (for assessment of endothelial dysfunction in patients with type I diabetes)—single study [47]
Optical coherence tomography angiography (OCTA)	A non-invasive imaging technique that uses motion contrast imaging to obtain information about volumetric blood flow [48]. It compares the decorrelation signal (differences in the intensity or amplitude of the backscattered OCT signal) between successive OCT scans taken in the same cross-section to construct a blood flow map [48].	Retinal vessels	83% (for assessing active myopic choroidal neovascularization)—single study [49] 75.7% (for assessing choroidal neovascularization)—single study [50] 98% (for detecting non-perfusion areas in eyes with diabetic retinopathy)—single study [51] 100% (for detecting retinal neovascularization in eyes with diabetic retinopathy)—single study [51] 83.7% (for differentiating patients without diabetic retinopathy form patients with non-proliferative diabetic retinopathy)—single study [52]	89% (for assessing active myopic choroidal neovascularization )—single study [49] 100% (for assessing choroidal neovascularization)—single study [50] 82% (for detecting non-perfusion areas in eyes with diabetic retinopathy)—single study [51] 100% (for detecting retinal neovascularization in eyes with diabetic retinopathy)—single study [51] 78.4% (for differentiating patients without diabetic retinopathy form patients with non-proliferative diabetic retinopathy)—single study [52]

**Table 2 diagnostics-12-02817-t002:** Pathophysiology of clinical complications associated with vascular aging.

Complications of Vascular Aging	Pathophysiology
Atherosclerotic vascular diseases	NTBIs accumulates in ECs and VSMCs, which undergo apoptosis. These cells produce elevated levels of VEGF and MCP1, which cause increased permeability and recruitment of immune cells [53]. Increased vascular permeability induces subendothelial infiltration of LDL, which promotes the development of atherosclerotic plaque [54]. In addition, impaired endothelial NO synthesis induced by iron overload promotes the activation of proatherogenic mechanisms, endothelial dysfunction, and arterial stiffness [55]. Under these conditions, monocytes recruited to the atherosclerotic plaque differentiate into M1 macrophages with a pro-inflammatory phenotype [56]. This phenotype is associated with poor cholesterol-carrying capacity and the formation of foam cells [57]. Increased necrosis of the core of the atherosclerotic plaque and its fibrosis is caused by apoptosis of VSMCs, formation of foam cells, and reduction in collagen content in the plaque. This promotes the development of CVD [58].
Hypertension	Vascular aging is characterized by structural changes such as increased intima-media thickness (IMT), vessel stiffening, and calcification [59]. With age, the proliferation and migration of VSMCs increases, while the elastic properties of the vessels decrease [60]. Vasorelaxation and vasoreactivity are also impaired due to impaired endothelial function. This ultimately increases the risk of developing hypertension [59].
Intracerebral hemorrhages	Iron accumulation in the brain triggers a cascade of harmful reactions, such as free radical production, mitochondrial damage, and macrophage/microglia activation, disrupting cellular homeostasis, which can lead to the development of ICH [61].
Aneurysms	The development of inflammation that occurs under oxidative stress leads to impaired endothelial function and apoptosis of ECs and VSCMs, resulting in increased vascular stiffness and increased vulnerability [62]. The persistence of oxidative stress results in the continuous activation of inflammatory pathways, which ultimately leads to the accumulation of proteoglycans, an increase in vessel volume in the vasa vasorum, and the degradation of elastic fibers, with a consequent change in the structure and composition of the vessel’s extracellular matrix (ECM) [63]. The consequence of this phenomenon is dissection, followed by or associated with aortic wall dilatation and rupture [63].
Vascular cognitive impairment (VCI)	NO is a critical vasodilatory factor that contributes to the regulation of blood flow in brain blood vessels, which can be inactivated by high levels of oxygen free radicals (ROS), resulting in significant cerebral vascular dysfunction [64]. In addition to regulating cerebral blood flow (CBF), NO also inhibits platelet aggregation and endothelial apoptosis and potentiates preservation of endothelial progenitor cells and anti-inflammatory properties [15]. All these benefits of NO are attenuated by ROS. Moreover, peroxynitrite, a product of NO oxidation, causes severe cytotoxicity, leading to cell death not only in the brain vasculature but also in other cell types, including neurons [64]. In addition, the severity of atherosclerotic lesions can lead to a reduction in CBF, which can also contribute to the development of VCI [65]. If the reduction in CBF is severe and persistent, strokes and myocardial infarction can occur [66].
Macular degeneration	With age, numerous changes occur within the retinal vasculature, such as reduced choroidal thickness and density, increased flow resistance, and, consequently, reduced choroidal flow [67]. In addition, decreased choroidal perfusion causes ischemia and exacerbates oxidative stress, which can lead to choroidal neovascularization [67]. Ultimately, these changes contribute to macular vascular degeneration [67].
Alzheimer’s disease	The development of inflammation that occurs under oxidative stress leads to impaired endothelial function and apoptosis of ECs and VSMCs, resulting in increased vascular stiffness and atherosclerosis of both large and small caliber vessels [68]. These changes, combined with the adverse effects of beta amyloid (Aβ), reduce cerebral perfusion and impair the ability of the cerebral circulation to deliver energy substrates and oxygen to active areas of the brain [68].

## Data Availability

Not applicable.

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
