# Peer review of "Vascular Aging and Damage in Patients with Iron Metabolism Disorders"

_diagnostics, 2022, doi:10.3390/diagnostics12112817_

Round 1
Reviewer 1 Report
The authors present a review on the current evidence of the association between impaired iron metabolism and vascular aging. They summarize current tools to investigate vascular aging, demonstrate the impact of different iron-related disorders, e.g. hereditary hemochromatosis or anemia on vascular outcomes, and present pathophysiological pathways. In particular they illustrate the current lack of research data on this topic.
The manuscript tackles an interesting and relevant topic, is well structured and reads well. However, I have major concerns about the article selection procedure; moreover some important discussion points are missing. Comments below.
* The description of the article selection procedure (Figure 1) is insufficiently described and not reproducible. Figure 1 states “Records excluded**” – but ** is not explained (I assume ** is a leftover from the original PRISMA flowchart template). Line 51f states articles were selected “based on the impact […] on current patient management.”, but this seems like a very subjective and arbitrary criterion – how was this decided in practice? PRISMA guidelines demand that the reasons for exclusion are explicitly given, so please provide the missing information.
* It is unclear which 86 articles from the reference list actually correspond to the ones “included in this review”. It is unclear if the 86 articles are original research work or if they include review articles. I suggest to give a table with an overview which articles belong to which of the presented sections.
* Section 8 “Iron deficiency” should probably be 7.2
* The “sensitivity” and “specificity” columns in Table 1 are misleading. These are partly based on reviews/meta-analysis and partly based on single study. Claiming generally applicable sensitivity and specificity value from one single study is difficult, since each study naturally has a specific sample profile that will influence the results (e.g. reference [31] studied only 4 men!) I suggest to transparently indicate in the table which values have been validated in several studies/are from meta-analyses and which ones are based on a single study.
* More emphasis should be placed on sex-specific results and the impact of menopause. Serum iron levels and stored body iron, e.g. hepatic iron are known to be different between men and women, and between pre- and postmenopausal women. are Menopause has an impact on both iron levels and vascular protection. This should be discussed further.
* What is currently missing from the manuscript is the link to type 2 diabetes. The link between iron overload and impaired glucose metabolism has already been shown, and the link between diabetes and micro- and macrovascular disease has also been shown. This definitely needs to be discussed as a potential pathway.
Reviewer 2 Report
Lines 179-180: “…. Undergo the Heber-Weiss reaction and then the Fenton reaction, …” It is “Haber-Weiss reaction”, not “Heber-Weiss reaction”. Also, Fenton reaction is a part of Haber-Weiss reaction, so “then” is not appropriate.
Lines 180-181: “… leading to the formation of oxygen free radicals, superoxides, and hydroxyl radicals.” Superoxide and hydroxyl radicals are two types of oxygen free radicals.
Line 182: What is “the peroxidation of proteins”?
Lines 180, 192 and 197 use the term “oxygen free radicals” and Line 188 uses the term “free oxygen radicals”. Please be consistent.
Lines 208 – 209: What do authors mean by “which due to its high pro-oxidant activity can generate ROS”. Superoxide is produced by the reduction of molecular oxygen, not because of high pro-oxidant activity.
Round 2
Reviewer 1 Report
I thank the authors for providing a revised version of their work. My comments were sufficiently adressed.
Reviewer 2 Report
No comments.